# Glutamine Synthetase as a Therapeutic Target for Cancer Treatment

**DOI:** 10.3390/ijms22041701

**Published:** 2021-02-08

**Authors:** Go Woon Kim, Dong Hoon Lee, Yu Hyun Jeon, Jung Yoo, So Yeon Kim, Sang Wu Lee, Ha Young Cho, So Hee Kwon

**Affiliations:** 1College of Pharmacy, Yonsei Institute of Pharmaceutical Sciences, Yonsei University, Incheon 21983, Korea; gowoon@yonsei.ac.kr (G.W.K.); tci30@naver.com (D.H.L.); hyun953@naver.com (Y.H.J.); jungy619@yonsei.ac.kr (J.Y.); ksy_dct@naver.com (S.Y.K.); tkddn407@naver.com (S.W.L.); hayoung.cho@yonsei.ac.kr (H.Y.C.); 2Department of Integrated OMICS for Biomedical Science, Yonsei University, Seoul 03722, Korea

**Keywords:** glutamine synthetase, glutamine metabolism, cancer metabolism, anticancer effect

## Abstract

The significance of glutamine in cancer metabolism has been extensively studied. Cancer cells consume an excessive amount of glutamine to facilitate rapid proliferation. Thus, glutamine depletion occurs in various cancer types, especially in poorly vascularized cancers. This makes glutamine synthetase (GS), the only enzyme responsible for de novo synthesizing glutamine, essential in cancer metabolism. In cancer, GS exhibits pro-tumoral features by synthesizing glutamine, supporting nucleotide synthesis. Furthermore, GS is highly expressed in the tumor microenvironment (TME) and provides glutamine to cancer cells, allowing cancer cells to maintain sufficient glutamine level for glutamine catabolism. Glutamine catabolism, the opposite reaction of glutamine synthesis by GS, is well known for supporting cancer cell proliferation via contributing biosynthesis of various essential molecules and energy production. Either glutamine anabolism or catabolism has a critical function in cancer metabolism depending on the complex nature and microenvironment of cancers. In this review, we focus on the role of GS in a variety of cancer types and microenvironments and highlight the mechanism of GS at the transcriptional and post-translational levels. Lastly, we discuss the therapeutic implications of targeting GS in cancer.

## 1. Introduction

Glutamine is the most abundant amino acid in the human body, constituting 20% of the total free amino acids in blood [1]. Cells acquire glutamine from circulation or de novo synthesis by glutamine synthetase (GS). GS is an enzyme converting glutamate and ammonia into glutamine using adenosine triphosphate (ATP). GS is particularly highly expressed in the liver, kidney, skeletal muscle, and brain. GS detoxifies ammonia in the liver [2] and regulates acid–base balance by controlling ammonium availability in the kidney [3]. In the skeletal muscle, glutamine synthesized by GS is consumed for energy-yielding pathways [4]. In the brain, GS is mostly expressed in astrocytes, controlling glutamate level to protect neurons from excitotoxicity [5].

Cancer cells rewire metabolism to facilitate rapid proliferation and maintain survival under harsh conditions, such as nutrient-deprived and poorly vascularized environments (Figure 1) [6,7]. Since glutamine provides both carbon and nitrogen for cellular biogenesis, glutamine is a favored resource for cancer metabolism. Glutamine supplies substrates for nucleotide, nonessential amino acid, nicotinamide adenine dinucleotide phosphate (NADPH), and glutathione (GSH) synthesis. Thus, glutamine provides the building block for macromolecules, regulates pH homeostasis, and controls redox balance. Likewise, glutamine replenishes α-ketoglutarate (α-KG) to the tricarboxylic acid (TCA) cycle after being catabolized, a process called anaplerosis [8,9].

In addition to the function of GS, the structure of GS is well understood. Human GS is a decamer composed of two pentameric rings which are held together mostly by van der Waals interactions and hydrogen bonds. There are 10 active sites in the GS decamer, each located in the interface of two faces of pentameric rings [10]. GS possesses β-grasp domain and glutamine synthetase catalytic domain (Figure 2). GS is encoded by glutamate-ammonia ligase (*GLUL*), which lies on chromosome 1q31.

Glutaminolysis is regarded as a hallmark of cancer metabolism and has been extensively studied [11,12]. Glutaminolysis is a process of degrading glutamine catalyzed by glutaminase (GLS), and this process is essential for generating energy in various cancer types. Many cancer cells rely on glutaminolysis and cannot survive in glutamine-deprived conditions as they are addicted to glutamine. Therefore, a strategy targeting glutaminolysis for cancer therapy has been proposed. For example, CB-839, a selective GLS1 inhibitor, has antitumor activity in various cancer types, and the clinical trial is ongoing in several solid tumors and hematological malignancies [13,14,15,16]. However, metabolic reprogramming is disparate depending on cancer subtypes. Some cancer types are independent of glutaminolysis for their tumorigenesis, showing resistance to glutaminolysis inhibition [17,18]. Moreover, most of the glutaminolysis-resistant cancer subtypes express high GS, which catalyzes the opposite reaction of glutaminolysis. Cancer cells expressing high GS are self-sufficient for glutamine and can survive in glutamine-deprived conditions. Emerging evidence indicates that GS is associated with cancer progression. Nonetheless, it is not clear how GS is associated with cancers and acts differently in various cancer types. In this review, we comprehensively summarize recent studies on GS organized by cancer types and tumor microenvironment (TME). In addition, we focus on the GS regulation mechanisms subdivided into transcriptional regulation and post-transcriptional regulation. Finally, we discuss the GS inhibitors and suggest future directions of GS investigations in cancer. 

## 2. Glutamine Synthetase and Cancer

### 2.1. Dysregulation of Glutamine Synthetase in Tumor

GS expression and functions are different for each cancer type, and this review focuses on specific cancer types in which GS plays a crucial role (Table 1).

#### 2.1.1. Glioma

Glioma cell growth does not depend on GLS, but it rather depends on GS [17,19]. Marin-Valencia et al. investigated glutamine metabolism in a human orthotopic tumor (HOT) model [19]. The glutamine/glutamate ratio is higher in the tumor compared to that of the normal brain. Consistently, glutamine per gram in tissue is increased in the tumor while glutamate is decreased. A human tissue array of 150 glioma specimens shows that GS expression is higher in glioblastoma (GBM), which is a high-grade glioma, than in low-grade glioma or oligodendroglioma (OD).

Unlike GS expression, GLS expression is higher in the surrounding brain tissues than in the tumor area [19]. Another anaplerosis-associated enzyme, pyruvate carboxylase (PC) is highly expressed in the tumor compared to the surrounding brain. PC converts pyruvate to oxaloacetate and contributes to the anaplerotic pathway of the TCA cycle. This is a consistent result with the previous report that showed glutamine-independent glioma cells have high PC activity and thus induce glucose-dependent anaplerosis, allowing cells to be glutamine-independent [20]. Not surprisingly, glutaminolysis inhibition in glioma does not affect cell proliferation [17]. Moreover, most of the glutamine required for the growth of GBM tumors is synthesized de novo [17]. The glutamine of GBM synthesized by GS is utilized for purine biosynthesis to support cell growth in glutamine-deprived conditions. These findings show that glutamine catabolism has minimal relevance in GBM tumorigenesis. Instead, GS upregulation in GBM supports tumorigenesis by supplying glutamine and enhancing nucleotide biosynthesis.

Several papers have reported that OD expresses low GS [19,21,22]. This characteristic of OD cells enables GS to be used as a diagnostic marker for distinguishing astrocytoma and OD, which have similar histological features [21]. GS is expressed in all grades of astrocytoma and oligoastrocytoma (total 19 tumor samples), whereas only 1 sample out of 16 samples expresses GS in OD. Since OD is a GS-negative tumor, the viability of OD cells decreases when glutamine is withdrawn [22]. Interestingly, unlike other GS-negative cancers, glutamine depletion does not decrease anaplerotic substrates, indicating OD cells do not use glutamine for anaplerosis. Instead, glutamine depletion induces nutritional stress, suppressing mTOR and Wnt/β-catenin pathways, which is mitigated by GS overexpression [22]. In addition, inhibition of glutamine uptake by its transporter (ASCT2, SNAT2, and LAT2) inhibitors blocks the proliferation of OD cells. Based on these results, OD cells exhibit unique metabolic characteristics, such that OD is glutamine auxotrophic but does not depend on glutamine anaplerosis for cell proliferation.

#### 2.1.2. Liver Cancer

GS is expressed distinctively in different types of liver tumors; several studies proposed that GS can be used as a diagnostic marker for distinguishing liver tumor types. In normal liver tissue, GS is expressed in pericentral hepatocytes, but it is not expressed in midzonal and periportal hepatocytes [23]. However, in hepatocellular carcinoma (HCC), the most common type of liver cancer, GS is strongly expressed in a diffused pattern [23]. GS can be used as a tracing marker for cell lineage in hepatocarcinogenesis. GS is highly expressed in HCC compared to cirrhosis and liver cell dysplasia [23]. Furthermore, analysis of GS expression in 260 liver tissue samples also showed that serum GS level is significantly increased in HCC patients compared to that of chronic hepatitis B stages 1 to 3 patients [24]. Furthermore, in HCC patients, GS level is positively associated with TNM staging and Child-Pugh score, which classifies the severity of cirrhosis. 

GS is highly expressed in hepatitis B virus (HBV)-related HCC compared to HBV alone samples [25]. Genome-wide association studies genotyping of 139 familial HBV-related HCC patients revealed that two single nucleotide polymorphism clusters are associated with familial HBV-related HCC patients [26]. One of them completely overlaps with the *GLUL* gene, and the other overlaps with SLC13A2, one of the glutamine transporters [26]. This study supports the differential expression of GS in HBV-related HCC.

In addition to HCC, a specific expression pattern of GS is observed in other liver tumor types. Hepatic adenoma (HA) expresses GS in the diffused pattern similar to HCC, and focal nodular hyperplasia (FNH) expresses GS in a heterogeneous map-like expression pattern [27,28]. The difference in GS expression facilitates distinction between HA and FNH, which are difficult to differentiate due to similarities in histological features. Indeed, the presence of map-like GS showed 85% sensitivity and 100% specificity in differentiating HA and FNH samples (n = 109) [27]. 

Liver cancer development and progression are associated with the altered β-catenin/Wnt pathway. The β-catenin-GS axis has been well studied in the liver tumor. *GLUL* gene is induced by activation of the β-catenin pathway in HCC and hepatoblastoma (HB) [29,30]. In addition, GS and β-catenin show a positive correlation in human HCC samples. When β-catenin was sequenced in 86 HCC samples, activating β-catenin mutations were only found in GS-positive HCC samples [31]. Array-comparative genomic hybridization showed that the genomic pattern of β-catenin is segregated according to the GS status [31]. To further elucidate the relationship between GS and β-catenin, Sanger sequencing of β-catenin exon 3 in HCC and HA samples was conducted [32]. A modest correlation existed between diffused GS immunostaining and β-catenin exon 3 mutation [32]. Moreover, a recent study reported that β-catenin mutation induces metabolic changes and autophagy, increasing the sensitivity of HCC cells to sorafenib by regulating GS expression [33]. β-catenin-GS axis also activates mTORC1, supporting liver cancer cell survival and growth [34]. These findings suggest that the β-catenin-GS axis is upregulated and has an important role in liver cancers. Furthermore, nuclear staining of β-catenin decreases after chemotherapy, whereas staining of GS still remains after chemotherapy [35]. Hence, GS would be useful for the assessment of section margin or identification of residual tumors after chemotherapy. 

The relationship between yes-associated protein 1 (YAP) and GS in liver cancer has also been studied [30,36]. YAP is a co-activator involved in the Hippo pathway. It regulates cell proliferation and organ size [37]. In liver cancer, YAP is frequently activated, and YAP-driven liver growth is susceptible to GS inhibition [36]. YAP induces the expression and activity of GS to reprogram nitrogen metabolism, including nucleotide biosynthesis, resulting in hepatomegaly and liver cancer cell growth. Furthermore, YAP1 is co-expressed with β-catenin, and YAP1/β-catenin mutants induce GS promoter activity in HB [30]. Together, these findings demonstrate that YAP cooperates with the anabolic demands of cell growth during tumorigenesis.

#### 2.1.3. Breast Cancer

Breast cancer cells carry differential glutamine dependency in a subtype-specific manner [18]. Basal type breast cancer cells express high GLS and low GS, yet luminal type breast cancer cells express high GS and low GLS, which was identified in both primary human breast cancer and breast cancer cell lines. GS expression determines glutamine dependency. Luminal type cells expressing high GS are glutamine-independent, whereas basal type cells are glutamine-dependent. However, when basal type cells and luminal type cells are co-cultured, basal type cells acquire glutamine independence through symbiosis with luminal cells. In line with this study, CB-839, a GLS inhibitor, has anti-proliferative activity in triple-negative breast cancer and basal-like HER2-positive breast cancer, but not in luminal type estrogen receptor-positive T47D cells [13].

GS is undetectable in normal breast tissues, but it is heterogeneously expressed in breast cancers, ranging from weak to extensively strong levels [45]. *GLUL* knockdown inhibits the proliferation of HER2-positive SK-BR-3 cells, which express higher GS than other types of breast cancer cells by blocking the ERK and p38 MAPK signaling pathways. Furthermore, there is a positive correlation between GS expression, tumor size, and HER2 level. In addition, overall survival (OS) and disease-free survival (DFS) are longer in GS-positive tumors than in GS-negative tumors.

#### 2.1.4. Ovarian Cancer

In ovarian cancer (OVC) cells, GS is heterogeneously expressed [38,40]. In OVC cells expressing high GS (GS^high^ OVC cells), *GLUL* silencing reduces cell proliferation via inhibition of the p38 MAPK signaling pathway [38]. Likewise, GS^high^ OVC cells depend on glucose rather than glutamine for anaplerosis to TCA cycle, and activated signal transducer and activator of transcription3 (STAT3) regulates glycolysis of GS^high^ OVC cells [39]. Therefore, co-targeting GS and STAT3 would be an effective strategy for treatment of GS^high^ OVC.

However, GS downregulation has been observed in 45 cases among 316 cases of the primary OVC [40]. The tumor growth is suppressed when GS is overexpressed in OVC cells expressing low GS (GS^low^ OVC cells) in vitro and in vivo. In line with this result, inhibition of the glutamine transporter ASCT2 is more effective in impeding proliferation of GS^low^ OVC cells than GS^high^ OVC cells. These findings suggest that glutaminolysis is more important in GS^low^ OVC cells than glutamine synthesis. Indeed, Yang et al. found that lowly invasive OVC cells are glutamine-independent, whereas highly invasive OVC cells are glutamine-dependent [39,46]. Such a result indicates glutaminolysis plays a more important role in OVC invasiveness than in glutamine synthesis, and GS does not play a critical role in tumorigenesis of GS^low^ OVC. Instead, GS expression is increased in cancer-associated fibroblasts (CAFs) to supply glutamine to OVC cells, supporting tumor growth [46].

#### 2.1.5. Lung Cancer

GS is expressed in the bronchial epithelium of normal lung tissues, and GS is highly expressed in lung tumors [41]. Although GLS1 is also increased in lung tumors, net glutamine is accumulated in lung tumors, indicating that increased GS overrides increased GLS1. In addition, *GLUL* mRNA level, but not *GLS*1, is increased in KRAS mutated non-small cell lung cancer (NSCLC) compared to wild-type [47]. Furthermore, as in GBM, NSCLC cells rely on glucose-derived anaplerosis rather than glutaminolysis-associated anaplerosis [48,49]. In addition, PC expression and activity are increased in NSCLC. PC knockdown decreases NSCLC cell proliferation, colony formation, and tumor growth in a mouse xenograft model [48]. These findings hint that GS may play an important role in lung cancer tumorigenesis.

Some reports demonstrate that GS is negatively associated with drug resistance in lung tumors [42,43]. GS is increased in gefitinib-sensitive lung cancer cells and decreased in gefitinib-resistant cells. GS overexpression in GS^low^ lung cancer cells re-sensitizes cells to gefitinib [42]. Conversely, GS knockout decreases sensitivity to gefitinib and induces metastasis. This phenomenon can be explained by the fact that GS overexpression reduces glutaminolysis, which in turn decreases energy production and GSH synthesis, resulting in drug sensitization. Similarly, Muthu et al. showed that GS ablation induces drug resistance by increasing the capacity of the malate-aspartate shuttle, which increases metabolic fitness, escaping drug pressure [43]. Although GS is highly expressed in lung cancer, its role in lung cancer is controversial, so further studies should be conducted to elucidate the exact role of GS in lung cancer.

#### 2.1.6. Pancreatic Cancer

Pancreatic ductal carcinoma (PDAC) has limited glutamine supply due to its highly fibrotic and poorly vascularized features [50]. GS expression is elevated in both pancreatic cancer patients and PDAC mouse models to overcome glutamine deficiency [44]. GLS inhibition exhibits anti-proliferative activity in cultured PDAC cells, while the antitumor effect has not been observed in PDAC mouse models [51]. In addition, GS is required for PDAC cells to be rescued by α-KG [44]. The GS-deficient PDAC cells still operate the TCA cycle, but they cannot support α-KG-mediated glutamine biosynthesis and subsequent nitrogen anabolism. Through these findings, it can be speculated that α-KG could be used for glutamine synthesis as well as anaplerosis in glutamine-starved PDAC. Indeed, α-KG receives nitrogen from various sources such as aspartate and alanine to synthesize glutamate, which is further converted into glutamine. *GLUL* knockout does not significantly affect oxygen consumption and ATP production, but nucleotide and hexosamine synthesis is inhibited. Moreover, *GLUL* ablation suppresses LSL-KrasG12D/+; Trp53f/f; Pdx1-Cre (KPC) tumor growth and increases the survival of mice [44]. These findings show that GS-coupled nitrogen metabolism is important for PDAC development and growth.

#### 2.1.7. Other Cancers

In sarcoma, GS is related to glutamine dependency. When L-asparaginase (ASNase), a glutamine depleting drug, is treated to human sarcoma cell lines, only a subset of cell lines are highly sensitive to ASNase. ASNase-resistant sarcoma cells highly express GS, and these cells become significantly sensitive to ASNase by treating GS inhibitor, methionine sulfoximine (MSO) [52]. Similarly, human pediatric sarcoma, rhabdomyosarcoma, and Ewing sarcoma cells adapt to glutamine depletion through GS expression [53]. GS supports nucleotide biosynthesis and optimal mitochondrial bioenergetics of sarcoma cells by providing glutamine. Therefore, GS inhibition with shRNA and chemical inhibitor reduces sarcoma cell proliferation in glutamine-deprived conditions, and tumor growth is significantly decreased in the orthotopic xenograft model.

There is a report showing that GS and ASCT2, glutamine transporter, are associated with gastric cancer (GC) [54]. GC cells have heterogeneous sensitivity to benzylserine (BenSer), which is an ASCT2 inhibitor. Since BenSer-resistant cells express GS, these cells are not affected by glutamine deprivation. GS inhibition in these cells markedly reduces GC cell proliferation. Furthermore, co-treatment of ASCT2 inhibitor and GS inhibitor significantly suppresses the tumor volume as compared to the single treatment of each drug.

The relationship between GS and radiation resistance has been reported in nasopharyngeal carcinoma (NPC) [55,56]. Radiation is a cancer treatment method that involves irradiating ionizing radiation to cancer tissue and causing DNA damage. Radiation-resistant cells possess high DNA damage repair capacity. Radiation-resistant NPC cells under radiation stress undergo metabolic reprogramming by increasing GS expression. Radiation-treated NPC cells obtain radiation resistance by increasing nucleotide synthesis for DNA repair [55]. The glutamine synthesized by GS is mainly used for de novo nucleotide synthesis and is used less for salvage pathway under radiation. GS knockdown in irradiated NPC cells increases basal respiration, mitochondrial ATP, basal glycolysis, and energy anaplerosis ability. These results demonstrate that GS knockdown reverses cell metabolism from radiation-induced nucleotide synthesis to increased glycolysis. In addition, GS promotes radiation-induced G2/M arrest recovery, and genetic ablation of GS re-sensitizes radiation-resistant NPC cells to radiation therapy in vivo [56]. These findings indicate that GS connects glutamine metabolism with radiotherapy response through modulation of nucleotide synthesis and DNA repair 

### 2.2. Dysregulation of Glutamine Synthetase in the Tumor Microenvironment

GS plays a crucial role in the TME as well as in cancer cells (Table 2). TME refers to the heterogeneous environments surrounding the cancer cells, consisting of the extracellular matrix, fibroblasts, adipose cells, immune cells, and blood vessels [57]. In the brain, GS is mostly expressed in astrocytes. However, GBM cells have highly heterogeneous GS expression as there are both GS^high^ GBM cells and GS^low^ GBM cells. GS^high^ GBM cells are self-sufficient for glutamine, whereas GS^low^ GBM cells rely on exogenous glutamine. Therefore, GS^low^ GBM cells maintain rapid cell proliferation by receiving glutamine from GS^high^ GBM cells or astrocytes [17,58].

In addition to astrocytes, GS plays a similar role in microglial cells [59]. Microglial cells are brain-resident macrophages and mostly remain in an inactive state. However, in pathological conditions, microglial cells are activated and gain inflammatory phenotype. GS inhibition increases the release of inflammatory mediators in vitro, and microglia-specific GS ablation increases inflammatory marker expressions in vivo. Consistent with this study, there is a report that GS is significantly upregulated in GBM tumor-associated macrophages (TAMs) compared to primary human microglia [60]. 

In OVC, the glutamine anabolic pathway is increased in CAFs compared to normal ovarian fibroblasts (NOFs) [46]. GS is upregulated in CAFs, supporting the survival of glutamine-addicted cancer cells by supplying glutamine. CAFs have higher metabolic flexibility than NOFs, so CAFs utilize carbon and nitrogen sources for glutamine synthesis through adaptive mechanisms. In glutamine-starved CAFs, more than 60% of the carbon sources of glutamine are contributed by glutamate. In addition, about 80% of the nitrogen sources for glutamine synthesis of CAFs are from ammonia and various sources such as branched-chain amino acids (BCAAs), aspartate, and alanine. On the other hand, NOFs use less than 10% of BCAAs, alanine, and ammonia as nitrogen sources to synthesize glutamine. These findings show that CAFs increases glutamine synthesis through dysregulated metabolism compared to NOFs. Moreover, CAFs support the increased glutamine catabolism of OVCs by providing external glutamine. Co-targeting stromal GS and GLS in OVCs synergistically reduces tumor growth in vivo.

In multiple myeloma (MM), GS is lowly expressed, and MM cells depend on extracellular glutamine [63,64]. On the contrary, GLS is highly expressed in MM, and MM cells are sensitive to GLS inhibition [63,65]. Since glutaminolysis is active in MM cells, MM cells consume large amounts of glutamine, causing glutamine depletion in bone marrow. It was observed that the differentiation of human mesenchymal stromal cells (MSCs) into osteoblasts is impaired by glutamine depletion of the bone marrow [66]. Interestingly, GS might be relevant to this impairment of differentiation induced by glutamine depletion [64]. In the bone marrow biopsies of MM patients, the GS expression in stromal cells is increased, and this result was also observed in MSCs co-cultured with MM cells [64]. Moreover, osteoblast markers, which are reduced by glutamine depletion, remain high when GS is silenced [64]. These findings imply that GS might have a certain role in microenvironment of MM. However, the exact function of GS has not yet been revealed in MM microenvironment, thus further research is necessary.

GS is involved in drug resistance in acute lymphoblastic leukemia (ALL) [62]. Adipocytes surrounding ALL cells protect ALL cells from ASNase, which is the first-line therapy for ALL. This is possible because GS is highly expressed in adipocytes, and GS provides glutamine to ALL cells. Indeed, in high-risk ALL patients, GS expression in adipocytes is increased after ASNase treatment. Similar to ALL, PDAC cells use glutamine produced by GS^high^ adipocyte, and cell proliferation is supported by GS^high^ adipocyte [67].

Recently, it has been reported that GS regulates the immune system by modulating macrophage differentiation. There are two types of macrophages: M1 and M2. M1 macrophages are typically activated macrophages, and M2 macrophages are alternatively activated macrophages. M2 macrophages, referred to as TAMs, have anti-inflammatory properties and promote tumor growth and metastasis. GS is an important factor in maintaining the M2 phenotype [61]. GS inhibition with MSO promotes the accumulation of succinate in M2 macrophages. Succinate is a crucial factor in regulating pro-inflammatory responses by inhibiting anti-inflammatory genes and stabilizing hypoxia inducible factor 1 α (HIF1α) [61]. As expected, MSO treatment increases the protein level and activity of HIF1α [61]. These findings indicate that GS maintains M2 phenotype by suppressing the accumulation of succinate and reducing HIF1α. Furthermore, GS inhibition induces the recruitment of lymphocytes and inhibits T cell suppression. Thus, GS inhibition makes macrophages lose M2 phenotype and gain M1 phenotype. GS inhibition also suppresses vascularization by regulating endothelial capillary formation and metastasis by modulating cancer cell motility. This finding suggests that GS acts as a critical modulator for immune cell function and can be a potential therapeutic target.

GS is highly expressed in endothelial cells and regulates vessel development [68]. GS inhibition suppresses angiogenesis of ocular and inflammatory skin disease without affecting healthy endothelial cells. This is attributed to the repression of endothelial cell migration by modulating RHOJ, which is an endothelial Rho-related GTP binding protein. However, the regulation of GS angiogenesis in cancer has not yet been studied. Since angiogenesis is a critical feature in cancer development, it is necessary to investigate whether GS controls cancer angiogenesis.

## 3. Regulation of GS

### 3.1. Transcriptional Regulation of GS

#### 3.1.1. c-Myc

Bott et al. revealed that conditional Myc overexpression increases *GLUL* mRNA by microarray analysis and identified how Myc regulates GS transcription [69]. Myc dimerizes with Max family and binds to the E box sequences, directly regulating the transcription of the target genes. *GLUL* gene promoter lacks Myc binding site, but it includes CpG island that can be demethylated by thymine-DNA glycosylase (TDG) [69]. Promoter analysis showed that Myc is recruited to the E boxes of TDG, increasing TDG expression and activity. In turn, TDG causes demethylation of *GLUL* promoter and induces expression. Moreover, Myc plays an important role in modulating cancer metabolism by upregulating GS. Myc-overexpressing MCF10A breast cancer cells show increased incorporation of 15N-NH4Cl into glutamine and asparagine. GS silencing inhibits such effect of Myc activation. Labeled ribonucleoside is not significantly increased in Myc-overexpressing MCF10A cells, whereas GS silencing markedly reduced the 15N incorporation into the ribonucleoside. This result suggests that nucleosides are rapidly consumed to synthesize DNA and RNA to maintain accelerated proliferation, even though GS promotes nucleoside synthesis. In line with this data, Myc-induced lung tumors possess high GS [41], and SF188 cells expressing high c-Myc exhibit high GS mRNA and protein levels [17]. These findings demonstrate that Myc indirectly upregulates GS transcription by demethylating *GLUL* promoter via TDG, thereby increasing the glutamine-dependent anabolic processes.

#### 3.1.2. β-Catenin

β-catenin is a well-known GS upstream regulator. The upregulation of GS by β-catenin was first revealed in the liver. GS expresses differentially in Wnt/β-catenin activated mouse liver, and the promoter of GS is activated by β-catenin [29,70]. Audard et al. reported that mutated β-catenin upregulates GS [71]. All different mutations of β-catenin in HCC increase the activity of the reporter gene containing GS 5′-enhancer [72]. Among them, the S45F mutant increases reporter activity to the greatest extent. Later, other studies observed that WT β-catenin also induces GS expression but to a lesser extent than mutated β-catenin [72,73].

#### 3.1.3. GATA Binding Protein 3 (GATA3)

As previously stated, the GS expression in breast cancer varies depending on the subtype [18]. GATA3, a master regulatory transcription factor in the differentiation of luminal type breast cancer (GS^high^ subtype), regulates the transcription of GS. Microarray analysis shows that GS is upregulated in GATA3 overexpressed mouse breast epithelial cells. Silencing of GATA3 reduces both GS mRNA and protein levels in MCF7, the luminal type cells. Conversely, overexpression of GATA3 induces the GS in MDA-MB-231, the basal type cells. ChIP analysis showed that GATA3 is enriched at -524 to -518 bp of GS promoter. These results reveal that GATA3 contributes to cell type-specific GS expression.

#### 3.1.4. Members of the Class O of Forkhead Box Transcription Factors (FOXO)

PI(3)K-PKB-FOXO signaling is a crucial pathway regulating cell proliferation, progression, and stress resistance [74]. Microarray results showed that FOXO3 regulates GS transcription [75]. FOXO3 binds to FOXO-responsive enhancer of GS at −2520 and −5000 bp as verified through ChIP and mutational analysis. FOXO-mediated GS upregulation causes mTOR inhibition by blocking the translocation of mTOR to the lysosome. As a result, upregulated GS by FOXO leads to increased autophagy, supporting cell survival.

#### 3.1.5. YAP1

*GLUL* mRNA is upregulated in YAP transgenic zebrafish liver, and YAP inhibition decreases GS expression [36]. ChIP analysis revealed that YAP is enriched in the transcriptional start site of the GS promoter in adult zebrafish liver. This finding was confirmed in the HCC cell line HepG2, indicating that YAP-induced GS upregulation is conserved in humans. Although promoter fragmentation analysis was performed, accurate YAP binding sites were not identified.

#### 3.1.6. Signal Transducer and Activator of Transcription 5 (STAT5)

GS expression is upregulated in radiation-resistant cells and promotes nucleotide synthesis to enable DNA repair, thereby maintaining cancer cell growth [55]. To identify the upstream regulator of GS under radiation, promoter analysis was conducted. As a result, STAT5-response elements in the GS promoter were identified. Indeed, STAT5 levels are increased in radiation-resistant cells. ChIP analysis and luciferase assay showed that STAT5 is recruited to two putative STAT5 binding sites in the *GLUL* promoter.

### 3.2. Posttranslational Modifications of GS

In glutamine-depleted condition, GS protein level is significantly induced, whereas the mRNA level is not changed [17]. This implies that GS can be modulated by posttranslational modification (PTM). Nguyen et al. showed that glutamine-dependent GS stability is regulated by acetylation (Figure 3) [76]. Under high glutamine, lysine 11 (K11) and K14 residues of GS are acetylated by histone acetyltransferases p300/CBP. Next, acetylated GS is recognized and ubiquitinated by cereblon (CRBN), an E3 ubiquitin ligase, resulting in proteasomal degradation. Later, it was revealed that valosin-containing protein (p97/VCP) promotes GS degradation by CRBN [77]. p97/VCP segregates large cellular structures to facilitate proteasomal degradation [78]. Ubiquitinated GS recruits p97/VCP which promotes degradation by disassembly of the homodecamer structure of GS into monomers [77]. In addition to CRBN, zinc and ring finger 1 (ZNRF1) was suggested as an E3 ligase regulating GS [79]. When nerve degeneration/regeneration occurs after nerve injury, the expression of GS in Schwann cells is regulated by ZNRF1-dependent proteasomal degradation, thereby regulating myelination and differentiation of Schwann cells.

GS is mostly located in the cytosol, but it also exists in the plasma membrane of endothelial cells [68]. Palmitoylation is a reversible PTM attaching fatty acids and contributes to the transfer of their targets to the plasma membrane [80]. Wan et al. first suggested the possibility of GS palmitoylation. By analyzing palmitoyl-proteome in Huntington’s disease mouse model, they found that GS is palmitoylated [81]. Palmitoylated GS was further confirmed using clickable palmitoylation probes [68]. Interestingly, MSO treatment decreases the palmitoylation of GS, indicating that the catalytic activity of GS affects its palmitoylation state. In addition, purified GS in a cell-free system palmitoylates itself by using palmitoyl-alkyne coenzyme A, substrate of palmitoyl moiety. These results show that GS auto-palmitoylates in endothelial cells. Auto-palmitoylated GS confers palmitoyl group to RHOJ, sustaining plasma membrane localization, and activity of RHOJ.

In addition to palmitoylation, γ-Aminobutyric Type B Receptors (GABABRs) regulate stability and localization of GS [82]. GABABRs are G protein-coupled receptors (GPCRs) expressed in mammalian astrocytes, consisting of R1 subunit and R2 subunit. R2 subunit binds to astrocytic GS and decreases proteasomal degradation of GS, thereby increasing stability and controlling its subcellular localization to the plasma membrane. However, the mechanism of GABABRs-associated GS regulation is not revealed clearly and needs to be further studied.

## 4. GS Inhibitors

GS inhibitors are well-reviewed by Eisenberg et al. and Berlicki et al. [83,84]. GS inhibitors can be divided into four groups: organosulfur analogues of glutamate, organophosphorus analogues of glutamate, bisphosphonate, and miscellaneous agents. Among these drugs, MSO, an organosulfur analogue of glutamate, is the most widely used compound in experiments. MSO is an irreversible competitive inhibitor of GS, and it effectively inhibits GS activity. However, MSO is not applicable in clinical practice due to its main adverse effect, convulsion. It has been reported that MSO-induced convulsion is alleviated by methionine treatment [85]. However, GS does not directly affect methionine. As a result, Brusilow and Peters have suggested the possibility that MSO-induced convulsion is not caused by targeting GS but by targeting methionine metabolism [86]. Moreover, because MSO cannot cross the blood–brain–barrier (BBB), MSO is not suitable for brain diseases. Therefore, the development of more selective and potent inhibitors targeting GS is inevitable.

## 5. Discussion

Recent studies have reported that glutamine is crucial for metabolic reprogramming in cancers. Glutamine catabolism, the conversion of glutamine to glutamate via glutaminolysis catalyzed by GLS, supports cancer cell proliferation by generating anti-oxidants and macromolecule biosynthesis. On the other hand, glutamine anabolism is also crucial in cancers as glutamine anabolism supports nucleotide synthesis by providing a nitrogen source (Table 1). Glutamine metabolism of cancers is regulated intricately by diverse factors. The factors affecting glutamine’s fate in cancers include tissue of origin, genetic mutation, presence of the drug, and culture methods. Since GS plays a vital role in tumorigenesis in particular cancer types, targeting GS could be considered as a potential therapeutic strategy. However, the decision of whether to target glutamine anabolism or catabolism should be made carefully after precisely characterizing the patients. GS plays a significant role not only in cancer but also in TME (Table 2). GS is upregulated in TME and supplies glutamine to cancer cells when glutamine is insufficient in cancer cells, especially in glutamine catabolism-activated cancer. Furthermore, GS controls angiogenesis by regulating the migration of endothelial cells [61,68]. As GS in TMEs supports cancer cell proliferation, targeting TME-specific GS would be an attractive option for cancer treatment.

It has been demonstrated that GS is transcriptionally regulated by a couple of oncogenic transcription regulators (Figure 3). GS expression is induced by β-catenin, FOXO, GATA3, c-Myc, STAT5, and YAP [18,29,36,55,69,70,74]. β-catenin, Myc, STAT5, and YAP are all well-known for their oncogenic roles [87,88,89,90,91]. However, FOXO, a transcription factor negatively regulated by PI(3)K-PKB/AKT signaling, is generally considered as a tumor suppressor. Nonetheless, FOXO is positively related to cancer progression in a context-dependent manner [92,93,94]. The role of FOXO-GS axis has not been studied in cancer. Therefore, further studies are necessary to identify the exact mechanism of GS regulation by FOXO. Furthermore, the specific promoter binding sites of these transcription regulators and the consensus sequence should be addressed.

Since the GS protein level is induced by glutamine deprivation without changing the mRNA level [17], it can be speculated that GS could be regulated at the PTM level. The only reported PTM related to GS stability is acetylation [76,77]. GS is acetylated by p300/CBP when glutamine is sufficient. Acetylated GS is recognized and ubiquitinated by CRBN. Next, GS is segregated by p97/VCP, undergoing proteasomal degradation. However, there are no studies connecting the epigenetic regulation of GS and GS-associated metabolism. Thus, it is pivotal to discover the underlying intertwined relationship between epigenetics and metabolism. In addition, there is still a lack of understanding of interacting partners or upstream regulators of GS, especially in the context of glutamine deprivation in cancer. If the role of the transcriptional or epigenetic regulators of GS in cancer is clearly identified, targeting GS by its regulators could be an effective therapeutic strategy. Furthermore, compensatory effects can occur as a result of targeting GS. Thus, it would be crucial to find out the major components that control the compensatory effects of GS inhibition. Co-targeting these components with GS could lead to synergistic effects. Collectively, GS exerts pro-tumoral features in particular cancer types and TMEs. Thus, targeting GS in precisely characterized and carefully selected patient groups based on individual metabolic profiles would be an efficient strategy for cancer treatment.

## Figures and Tables

**Figure 1 ijms-22-01701-f001:**
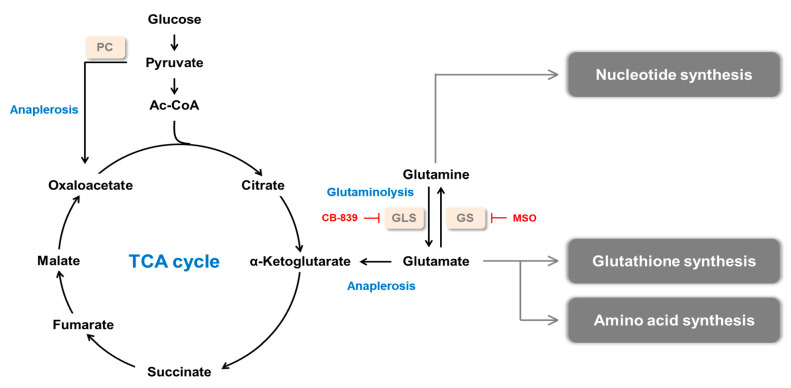
Glutamine metabolism in cancer. Glutamine synthesized by GS contributes to nucleotide synthesis. Glutamine is converted to glutamate by GLS, and glutamate is used for the biosynthesis of glutathione (GSH) and amino acid. α-ketoglutarate (α-KG) is converted from glutamate and replenishes tricarboxylic acid (TCA) cycle, supporting the synthesis of diverse macromolecules. PC: pyruvate carboxylase, GLS: glutaminase, GS: glutamine synthetase, MSO: methionine sulfoximine.

**Figure 2 ijms-22-01701-f002:**
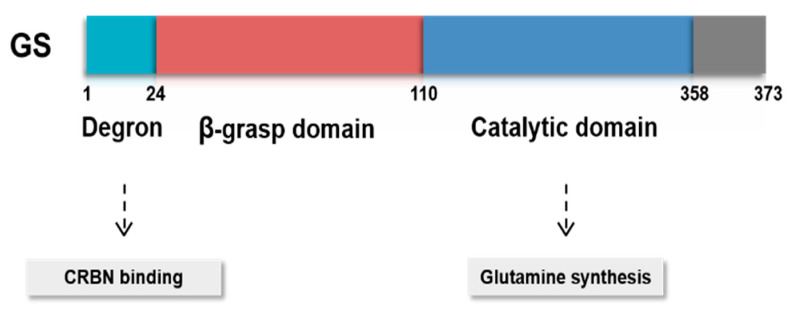
Functional domains of GS. GS possesses β-grasp domain and glutamine synthetase catalytic domain. Cereblon (CRBN) binds to GS degron, regulating its protein stability under sufficient glutamine level.

**Figure 3 ijms-22-01701-f003:**
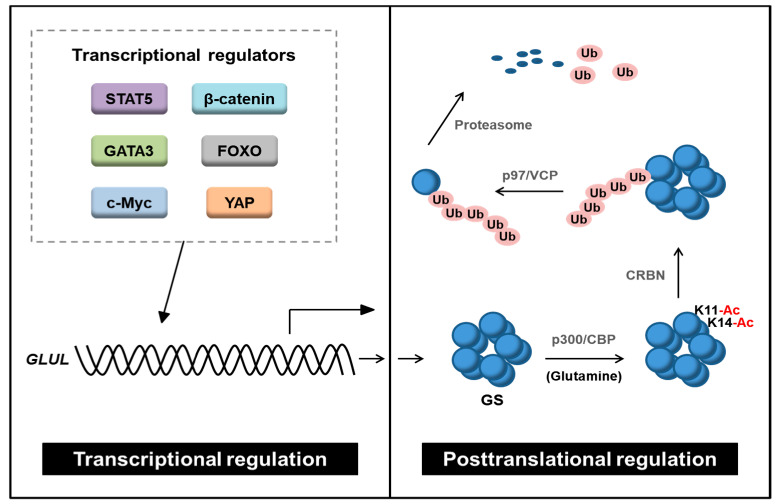
Transcriptional and posttranslational regulation of GS. GS is transcriptionally regulated by β-catenin, STAT5, GATA3, FOXO, YAP, and c-Myc. GS protein stability is regulated by acetylation. GS is acetylated by histone acetyltransferase, p300/CBP, and acetylated GS is ubiquitinated by the E3 ligase, CRBN. Ubiquitinated GS is segregated by p97/VCP and then degraded by the proteasome. STAT5: signal transducer and activator of transcription 5, GATA3: GATA Binding Protein 3, YAP: yes-associated protein, FOXO: members of the class O of forkhead box transcription factors, GS: glutamine synthetase, CBP: cyclic AMP response element-binding protein, CRBN: cereblon, Ub: ubiquitin, p97/VCP: valosin-containing protein.

**Table 1 ijms-22-01701-t001:** Diverse roles of GS in different cancer types.

Cancer Type	GS Expression	Role of GS	Experimental Models	References
GBM	High	GS sustains nucleotide biosynthesis and cell growth of GBM in gln starved conditions	Human GBM patients, GBM PDX model	[17,19]
OVC	High	GS supports the proliferation of OVC cells GS^high^ OVC shows low invasiveness	OVC cells	[38,39]
	Low	GS^low^ OVC shows high invasiveness	Xenograft mouse model	[39,40]
Breast cancer (luminal)	High	High expression of GS contributes to gln independence GLS inhibitor has no anti-proliferative activity	Luminal type breast cancer cells	[13,18]
Breast cancer (basal)	Low	Low expression of GS contributes to gln dependence GLS inhibitor has anti-proliferative activity	Basal type breast cancer cells, Xenograft model of basal like breast cancer	[13,18]
Lung cancer	High	Increased GS accumulates gln in cancer cells although gln catabolism is activated	GEMs (Myc-induced lung tumors)	[41]
	-	GS confers gefitinib resistance	NSCLC cells	[42,43]
PDAC	High	GS contributes to cataplerotic usage of α-KG *GLUL* ablation suppresses tumor growth	KPC tumor cell organoids, Orthotopic mouse model	[44]

**Table 2 ijms-22-01701-t002:** Diverse roles of GS in TMEs.

TME Cell Type	GS Expression	Role of GS	Experimental Models	References
TAMs	High	GS maintains M2 macrophage phenotype by suppressing the accumulation of succinate and HIF1α GS supports vascularization and metastasis of cancer cells	Lewis lung carcinoma implanted *GLUL* conditional knockout mice	[61]
Microglial cells	High	GS modulates inflammatory responses GS ablation in microglia increases inflammatory responses	Microglial-specific *GLUL* conditional knockout mice, Experimental autoimmune encephalomyelitis	[60]
GBM astrocytes	High	Astrocytes synthesize gln via GS and provide gln to GS^low^ GBM cells, supporting cell proliferation	Co-culture of rat primary cortical astrocytes and GBM cells	[17]
OVC CAFs	High	GS supports gln catabolism in OVC cells via crosstalk between CAFs and OVC Co-targeting of stromal GS and cancer GLS significantly suppresses tumor growth	Orthotopic mouse model	[46]
ALL adipocytes	High	GS protects ALL cells from L-asparaginase by supplying gln	Co-culture of leukemic cells with adipocytes, Leukemic mouse model	[62]

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
