# Peer review of "Glutamine Synthetase as a Therapeutic Target for Cancer Treatment"

_ijms, 2021, doi:10.3390/ijms22041701_

Round 1

Reviewer 1 Report

The review describes the role of glutamine synthetase in cancer and its possible role as therapeutic target. although the review is well written, it lacks the description of glutamine synthetase in multiple myleoma. Some recent papers have highlighted the involvement of glutamine metabolism in the pathogenesis of multiple myeloma. Over the recent years, some inhibitors of gln enzymes (such as GLS) have been tested in vitro and in vivo in myeloma models. 

Morever, in the microenvironment section the authors did not mention the research papers, recently published, describing the involvement of gln metabolism in multiple myeloma microenovironement (osteoblast, mesenchymal, skeletal cells etc.

 All these aspect must be discussed and added in the table if appropriate.

All these comments will improve the review 

Author Response

The review describes the role of glutamine synthetase in cancer and its possible role as therapeutic target. although the review is well written, it lacks the description of glutamine synthetase in multiple myleoma. Some recent papers have highlighted the involvement of glutamine metabolism in the pathogenesis of multiple myeloma. Over the recent years, some inhibitors of gln enzymes (such as GLS) have been tested in vitro and in vivo in myeloma models. 

Moreover, in the microenvironment section the authors did not mention the research papers, recently published, describing the involvement of gln metabolism in multiple myeloma microenovironement (osteoblast, mesenchymal, skeletal cells etc).

All these aspects must be discussed and added in the table if appropriate.

All these comments will improve the review. 

We thank the referee for recommending the way to improve this paper. As you have suggested, we added the section describing multiple myeloma (MM) in tumor microenvironment part as below in line 266-275. We wrote that glutaminolysis is active in multiple myeloma, and GLS inhibitor has antitumor activity in multiple myeloma. We did not add this content to the tumor part (section 2.1) because the focus of this study is on GS, not GLS. We placed this content in microenvironment part as there is a report that proposed the possible importance of GS in MM microenvironment. However, the role of GS in tumor microenvironment has not yet been clarified, so we did not add this content in the table.

In multiple myeloma (MM), GS is lowly expressed, and MM cells depend on extracellular glutamine. On the contrary, GLS is highly expressed in MM, and MM cells are sensitive to GLS inhibition. Since glutaminolysis is active in MM cells, MM cells consume large amounts of glutamine, causing glutamine depletion in bone marrow. It was observed that the differentiation of human mesenchymal stromal cells (MSCs) into osteoblasts is impaired by glutamine depletion of the bone marrow. Interestingly, GS might be relevant to this impairment of differentiation induced by glutamine depletion. In the bone marrow biopsies of MM patients, the GS expression in stromal cells is increased, and this result was also observed in MSCs co-cultured with MM cells. Moreover, osteoblast markers, which are reduced by glutamine depletion, remain high when GS is silenced. These findings imply that GS might have a certain role in microenvironment of MM. However, the exact function of GS has not yet been revealed in MM microenvironment, thus further research is necessary.

Reviewer 2 Report

The review by Kim et al. concerns the potential role of Glutamine Synthetase as a therapeutic target in cancer. The subject is interesting: GS role in several types of cancers is progressively appreciated, yet still awaiting an in depth comprehension. Literature in the field is rapidly expanding, and this renders the review particularly useful. The description of GS role in cancer and of its regulation is overall adequate. However, I have some critical remarks.

1) The statement at lines 53-54 seems of common sense but there is at least one important exception. In Chiu et al. (Br J Cancer. 2014 Sep 9;111(6):1159-67.) it is clearly demonstrated that HepG2, a well known and widely used beta-catenin mutated line derived from liver cancer, expresses high levels of GS although it is clearly sensitive to glutamine depletion. Interestingly, cancer cell growth is significantly inhibited by glutamine depletion, both in vitro and in vivo, and completely suppressed if glutamine depletion is associated with the  inhibition of GS with MSO.  

2) Again on liver cancer, hepatoblastoma should be cited as another example of liver cancer in which GS expression is driven by beta-catenin and YAP (see, for example, the recent contribution by Min et al., Am. J. Pathol. 2019, 189, 1091-1104).

3) Table 1 should include liver cancers; "type" should be modified in "types" in the caption of the Table.  

4) Besides basal-type breast cancer and oligodendroglioma, multiple myeloma is also GS negative (Bolzoni et al., Blood. 2016 Aug 4;128(5):667-79), uses large amounts of the amino acid, and is extremely sensitive to glutamine depletion (Soncini et al., Blood Adv. 2020 Sep 22;4(18):4312-4326). As a consequence of myeloma-derived glutamine depletion, GS is overexpressed in stromal bone marrow cells of myeloma patients  (Chiu et al., Cancers (Basel), 2020 12:3267).

5) The mechanism proposed by Ehsanipour et al. (Ref. 60) should not be generalized. Indeed, as shown in Fig. 5 of that reference, sensitivity to Gln depletion (and, hence, protection by stromal GS) is mainly seen in ALL cells with a sizable expression of Asparagine Synthetase (ASNS), while in ASNS-negative cells the effect is much more modest.  

6) Line 117. I have not been able to find a reference demonstrating that the dicarboxylic acid carrier coded by SLC13A2 transports also glutamine, as stated by the Authors. Please, check this issue and, if possible, add the reference. Alternatively, the sentence should be amended.  

Author Response

We appreciate the reviewers for recognizing the potential importance of our findings. We feel our responses to his/her criticisms have greatly improved the manuscript. Our responses to the reviewer’s queries are described point-by-point below.

The review by Kim et al. concerns the potential role of Glutamine Synthetase as a therapeutic target in cancer. The subject is interesting: GS role in several types of cancers is progressively appreciated, yet still awaiting an in depth comprehension. Literature in the field is rapidly expanding, and this renders the review particularly useful. The description of GS role in cancer and of its regulation is overall adequate. However, I have some critical remarks.

1) The statement at lines 53-54 seems of common sense but there is at least one important exception. In Chiu et al. (Br J Cancer. 2014 Sep 9;111(6):1159-67.) it is clearly demonstrated that HepG2, a well known and widely used beta-catenin mutated line derived from liver cancer, expresses high levels of GS although it is clearly sensitive to glutamine depletion. Interestingly, cancer cell growth is significantly inhibited by glutamine depletion, both in vitro and in vivo, and completely suppressed if glutamine depletion is associated with the inhibition of GS with MSO.  

Thank you for the comments. As for the line 53-54, we wrote this statement based on generally reported results, so we did not add detailed descriptions about exceptions. However, we totally agree with your idea. Therefore, we changed the statement at lines 53-54 from “glutaminolysis-resistant cancer subtypes express high GS” to “most of the glutaminolysis-resistant cancer subtypes express high GS” acknowledging the presence of exceptions.

2) Again on liver cancer, hepatoblastoma should be cited as another example of liver cancer in which GS expression is driven by beta-catenin and YAP (see, for example, the recent contribution by Min et al., Am. J. Pathol. 2019, 189, 1091-1104).

As the referee suggested, we added hepatoblastoma as an additional example (reference 30, Line 225, 241-242).

3) Table 1 should include liver cancers; "type" should be modified in "types" in the caption of the Table.  

Thank you for this comment. We corrected this error in table 1.

4) Besides basal-type breast cancer and oligodendroglioma, multiple myeloma is also GS negative (Bolzoni et al., Blood. 2016 Aug 4;128(5):667-79), uses large amounts of the amino acid, and is extremely sensitive to glutamine depletion (Soncini et al., Blood Adv. 2020 Sep 22;4(18):4312-4326). As a consequence of myeloma-derived glutamine depletion, GS is overexpressed in stromal bone marrow cells of myeloma patients  (Chiu et al., Cancers (Basel), 2020 12:3267).

According to the referee's opinion, we added the section describing multiple myeloma in tumor microenvironment part as below in line 266-275.

In multiple myeloma, GS is lowly expressed, and MM cells depend on extracellular glutamine. On the contrary, GLS is highly expressed in MM, and MM cells are sensitive to GLS inhibition. Since glutaminolysis is active in MM cells, MM cells consume large amounts of glutamine, causing glutamine depletion in bone marrow. It was observed that the differentiation of human mesenchymal stromal cells (MSCs) into osteoblasts is impaired by glutamine depletion of the bone marrow. Interestingly, GS might be relevant to this impairment of differentiation induced by glutamine depletion. In the bone marrow biopsies of MM patients, the GS expression in stromal cells is increased, and this result was also observed in MSCs co-cultured with MM cells. Moreover, osteoblast markers, which are reduced by glutamine depletion, remain high when GS is silenced. These findings imply that GS might have a certain role in microenvironment of MM. However, the exact function of GS has not yet been revealed in MM microenvironment, thus further research is necessary.

5) The mechanism proposed by Ehsanipour et al. (Ref. 60) should not be generalized. Indeed, as shown in Fig. 5 of that reference, sensitivity to Gln depletion (and, hence, protection by stromal GS) is mainly seen in ALL cells with a sizable expression of Asparagine Synthetase (ASNS), while in ASNS-negative cells the effect is much more modest.  

We thank the referee for this careful comment. We checked the Fig.5 in Ehsanipour et al. (Ref. 60) that the referee mentioned. According to the authors in this article, the sensitivity to both amino acids depletion (glutamine and asparagine) could not be explained by ASNS and GS expression. This finding is contrary to other papers that proposed GS expression negatively correlates with sensitivity to glutamine depletion.

However, this result is not their main idea. Fig.5 does not weaken their main finding that adipocytes expressing high GS provide glutamine to leukemia cells. In addition, our manuscript focuses on the microenvironment of ALL, not ALL itself. Moreover, the paragraph (line 276-280) in our review did not mention about the findings from Fig.5. Therefore, we believe there is no problem with the idea of the paragraph, and we did not change the contents of these findings in the paper.

6) Line 117. I have not been able to find a reference demonstrating that the dicarboxylic acid carrier coded by SLC13A2 transports also glutamine, as stated by the Authors. Please, check this issue and, if possible, add the reference. Alternatively, the sentence should be amended.  

We are sorry for the confusion caused by the reference issue. The reference for statement at line 117 (116 in revised manuscript) is the same reference in the previous sentence (reference 26). We added the reference to this sentence.

Reviewer 3 Report

This is a well written review article that extensively addresses the importance of GS in cancer.

Minor revisions:

  1. Please provide a list of abbreviations
  2. line 51, change metabolism reprogramming to metabolic reprogramming
  3. line 79, change human orthotropic tumors (HOTs) model to tumor model (no S).
  4. TNM full form?
  5. HBV full form?
  6. In pancreatic cancer, any evidence of GS activity in stellate cells?
  7. Table 2, column 1 heading, has to be TME cell type

Author Response

We appreciate the reviewers for recognizing the potential importance of our findings. We feel our responses to his/her criticisms have greatly improved the manuscript. Our responses to the reviewer’s queries are described point-by-point below.

This is a well written review article that extensively addresses the importance of GS in cancer.

Minor revisions:

  1. Please provide a list of abbreviations

We thank the referee for recommending the way to improve this review. We added the abbreviations in line 445.

  1. line 51, change metabolism reprogramming to metabolic reprogramming

Thank you for this comment. In line 51, we corrected the error as referee’s recommendation.

  1. line 79, change human orthotropic tumors (HOTs) model to tumor model (no S).

Thank you for this comment. In line 79, we corrected the error as referee’s recommendation.

  1. TNM full form?

Thank you for this comment. However, ‘TNM staging’ (or ‘TNM classification’) is full form itself so I did not change this word.

  1. HBV full form?

As reviewer’s suggestion, we added the full form of HBV (hepatitis B virus) in line 113, which is the first sentence stating HBV in this review.

  1. In pancreatic cancer, any evidence of GS activity in stellate cells?

There is no study directly estimating GS activity in pancreatic stellate cells (PSC). Nonetheless, there is a study that could predict GS activity in PSC (Sousa et al., nature. 2016 August 25; 536(7617): 479-483). This research investigated the deregulated metabolism in PSCs. PSCs have critical role in PDAC metabolism by providing non-essential amino acids (NEAA). However, analysis of metabolites levels in media secreted by human PSCs showed that the level of glutamine was low, whereas that of alanine was high. Therefore, the authors of this research suggested that alanine outcompetes glucose and glutamine as a carbon source to replenish TCA cycle. Based on this research, we could expect that GS activity would be low in PSCs. However, the role of GS should be further studied in PSCs to clearly understand the metabolism of pancreatic cancer. 

  1. Table 2, column 1 heading, has to be TME cell type

Thank you for your comment. We changed the column 1 heading to TME cell type as reviewer’s recommend.

Round 2

Reviewer 1 Report

I accept the paper in present form